# The CAM Model—Q&A with Experts

**DOI:** 10.3390/cancers15010191

**Published:** 2022-12-28

**Authors:** Dagmar Fischer, Georg Fluegen, Paul Garcia, Nassim Ghaffari-Tabrizi-Wizsy, Laura Gribaldo, Ruby Yun-Ju Huang, Volker Rasche, Domenico Ribatti, Xavier Rousset, Marta Texeira Pinto, Jean Viallet, Yan Wang, Regine Schneider-Stock

**Affiliations:** 1Division of Pharmaceutical Technology, Department of Chemistry and Pharmacy, Friedrich-Alexander-Universität Erlangen-Nürnberg, 91058 Erlangen, Germany; 2Department of General, Visceral, Thoracic and Pediatric Surgery (A), Medical Faculty, Heinrich-Heine-University, University Hospital Duesseldorf, 40225 Duesseldorf, Germany; 3Institute for Advanced Biosciences, Research Center Université Grenoble Alpes (UGA)/Inserm U 1209/CNRS 5309, 38700 La Tronche, France; 4R&D Department, Inovotion, 38700 La Tronche, France; 5SFL Chicken CAM Lab, Department of Immunology, Otto Loewi Research Center, Medical University of Graz, 8010 Graz, Austria; 6European Commission, Joint Research Centre (JRC), 21027 Ispra, Italy; 7School of Medicine, College of Medicine, National Taiwan University, Taipei 10051, Taiwan; 8Graduate Institute of Oncology, College of Medicine, National Taiwan University, Taipei 10051, Taiwan; 9Department of Obstetrics & Gynaecology, Yong Loo Lin School of Medicine, National University of Singapore, Singapore 119077, Singapore; 10Department of Internal Medicine II, Ulm University Medical Center, 89073 Ulm, Germany; 11Department of Translational Biomedicine and Neurosciences, University of Bari “Aldo Moro”, 70124 Bari, Italy; 12i3S—Instituto de Investigação e Inovação em Saúde, Universidade do Porto, 4200-135 Porto, Portugal; 13Ipatimup—Instituto de Patologia e Imunologia Molecular da Universidade do Porto, 4200-135 Porto, Portugal; 14Experimental Tumorpathology, Institute of Pathology, Universitätsklinikum Erlangen, FAU Erlangen-Nürnberg, 91054 Erlangen, Germany; 15Comprehensive Cancer Center Erlangen-EMN (CCC ER-EMN), Universitätsklinikum Erlangen, FAU Erlangen-Nürnberg, 94054 Erlangen, Germany

**Keywords:** tumor growth, angiogenesis model, metastasis model, standard operation procedures, immune-oncology, drug discovery, personalized therapy

## Abstract

**Simple Summary:**

The chorioallantoic membrane (CAM) model is an alternative in vivo test system for the study of hallmarks of cancer such as angiogenesis, tumor growth, immune escape, metastasis, and drug resistance. It is suitable for tumor cell lines and patient-derived xenografts (PDX), allowing for studying treatment protocols for cancer patients. The purpose of this review is to give answers to the most challenging questions around the CAM model, which is not only bridging in vitro and mouse in vivo studies but also has an interesting role in itself.

**Abstract:**

The chick chorioallantoic membrane (CAM), as an extraembryonic tissue layer generated by the fusion of the chorion with the vascularized allantoic membrane, is easily accessible for manipulation. Indeed, grafting tumor cells on the CAM lets xenografts/ovografts develop in a few days for further investigations. Thus, the CAM model represents an alternative test system that is a simple, fast, and low-cost tool to study tumor growth, drug response, or angiogenesis in vivo. Recently, a new era for the CAM model in immune-oncology-based drug discovery has been opened up. Although there are many advantages offering extraordinary and unique applications in cancer research, it has also disadvantages and limitations. This review will discuss the pros and cons with experts in the field.

## 1. Introduction

Recently, the chorioallantoic membrane (CAM) model raised considerable attention in the field of tumor biology, imaging, and cytotoxicity research, as it provides an attractive alternative model with respect to the 3R guidelines [1,2,3]. In this review, we will discuss the most important topics that could be relevant for a new but also an experienced researcher in this field, providing both more general and specific information about this in vivo model. A panel of experts answers the most challenging questions in an interview format.

### We Start with a Simple Question—What Is the CAM?

The allantois of the chick embryo appears at about embryonic day (ED) 3.5 as an evagination from the ventral wall of the endodermal hind gut. It pushes out of the body of the embryo into the extraembryonic coelom. The allantoic vesicle enlarges very rapidly and, during this process, the mesodermal layer of the allantois fuses with the adjacent mesodermal layer of the chorion, two extraembryonic tissues, to form the chorioallantoic membrane (CAM) [4]. In this double layer, an extremely rich vascular network develops, connecting to the embryonic circulation allantoic arteries and vein. The CAM vascularization undergoes three phases of development characterized by sprouting and intussusceptive microvascular growth (IMG [5]). In the early phase, multiple capillary sprouts invade the mesenchyme, fuse, and form the primary capillary plexus. During the second phase, sprouts are no longer present and have been replaced by a tissue-pillar expression of IMG. During the late phase, the growing pillars increase in size to form intercapillary meshes. Immature blood vessels scattered in the mesoderm grow very rapidly until ED 8 and give rise to a capillary plexus, which comes to be intimately associated with the overlying chorionic epithelial cells and mediates gas exchange with the outer environment. Rapid capillary proliferation continues until ED 11. At ED 14, the capillary plexus is located at the surface of the ectoderm adjacent to the shell membrane. Thereafter, the endothelial cell mitotic index declines rapidly, and the vascular system attains its final arrangement on ED 18, just before hatching [6]. 

At the beginning of the development of the chick embryo, before ED 6, the CAM performs a gas exchange function. Later, the rich vascularization and the position of the allantois immediately subjacent to the porous shell confer a respiratory function on the highly vascularized CAM. In addition to the respiratory function, the allantois also serves as a reservoir for the waste products excreted by the embryo, mostly urea at first and chiefly uric acid later.

## 2. Regularities

### 2.1. What Are the Major General Prerequisites for Alternative Animal Test System? 

At the European level, Directive 2010/63 on the protection of animals used for scientific purposes aims to ensure animal protection and adequate care for animals [7]. The use of animals for scientific or educational purposes should be considered only when there is no other alternative and it is governed by the principles of replacement, reduction, and refinement (3R). Animals should be replaced by less sentient alternatives such as invertebrates or in vitro methods whenever possible. Availability and awareness of adequate in vitro techniques represent the prerequisites for the use of alternative methods. The chosen methods must use the least number of animals, provide satisfactory results, use the species with the least ability to experience pain, suffering, anguish, and damage, and be optimal for the extrapolation of results to target species such as humans. In regulatory toxicology, the method must fulfil the phases of pre-validation (previous inter-laboratory assessment), validation of its reproducibility and relevance to in vivo toxicity, the independent assessment of the study by a panel of experts, and the progression toward regulatory acceptance.

Therefore, the chick embryo model has the main prerequisites to be considered as an alternative method to animal (rodent) experimentation bridging in vitro and mouse in vivo studies (Figure 1). It fulfills the 3R principle for a reduction in animal experiments as a kind of pre-orientation and strategic refinement for future necessary experiments. The careful evaluation of CAM xenografts provides initially important information about the targeted cellular functions of specific gene knockouts or drug treatments to develop hypotheses and to plan further suitable approaches. Nevertheless, the CAM model has an interesting role in itself that is highlighted in the following paragraphs.

### 2.2. What Is Needed to Run a CAM Lab?

Supervising a CAM lab is manageable, simple, and cost-efficient, particularly in comparison to rodent animal models. What you need is an egg incubator that provides the eggs with a constant temperature level and the right amount of humidity. Automatic incubators with rolling trays are available on the market. A stereomicroscope designed for low-magnification observation of a sample is needed. When you open a CAM lab, it is necessary to define standard operation procedures (SOPs). In these written instructions, the single steps to be performed during your CAM experiment, including the waste management and safety precautions, have to be described in detail. It is also very important to include information about potential hazards and how these hazards will be mitigated. The user must be aware that potential hazards are handled that are possibly classified as biohazards level 1 or 2.

This is the case when working with genetically modified organisms, CRISPR Cas-modified cell lines, sh-transfected cells, etc. This biohazardous material needs to be autoclaved or needs to be burned. It is also necessary to state how the chicken embryo must be sacrificed after the experiment. Every new lab member using the CAM model needs to be instructed carefully based on these SOPs.

## 3. Tumor Growth Features on the CAM

### 3.1. How Can a CAM Tumor Graft Be Exploited?

The CAM model is an excellent tool to study several hallmarks of cancer [8], which describe the major biological capabilities of tumors considering their complexity and heterogeneity. According to this policy paper, there are eight core hallmarks supplemented by so-called enabling characteristics that allow tumor cells to execute these features (Figure 2). Several applications of the chicken CAM in the frame of cancer-hallmark-related studies are discussed [9,10], and more details are given in the following paragraphs. Mostly, cancer cells or patient-derived tissues are transplanted onto the CAM, where within 2–10 days tumor formation occurs. After harvesting these tumor xenografts (ovografts), they could be formalin-fixed and embedded in paraffin and, thus, will be available as tumor blocks for immunohistochemical staining with antibodies specific to key markers involved in tumor progression, proliferation, tumor-induced angiogenesis, tumor cell intravasation, apoptosis, metabolism, or drug response (Figure 2). The ovografts can also be analyzed by flow cytometry or immunofluorescence. Moreover, the tumor cell induced remodeling of the collagen-rich CAM matrix as signs of interaction with the tumor microenvironment can be studied using different imaging techniques [11].

### 3.2. How Can CAM Xenografts Be Used to Study Cell Death?

Besides from tumor-induced angiogenesis, the CAM can be used to study the ability of tumor cells to resist cell death, to enable replicative immortality, and to evade growth suppressors. Depending on the cell line used, necrosis on CAM-xenografted tumors can be observed macroscopically, through the shell window or after tumor/CAM excision. As an example, the co-existence of angiogenesis, necrosis, and hemorrhage is pathognomonic for human glioblastoma multiforme (GBM) observed in CAM-xenografted GBM tumors [12]. CAM-derived biologic material can be processed as human samples; thus, histological staining of tumor slides can provide relevant information regarding the presence of necrotic tissue and cells with apoptotic phenotypes. Furthermore, apoptosis and tumor cell proliferation can be addressed within several contexts such as the study of the effects of radiotherapy. Excised CAM tumors, developing from cells or tissues subjected to radiation schedules, constitute the ideal biologic material to measure cell viability, proliferation, and death via respective assays (e.g., Ki-67, WST-1, TUNEL, etc.), flow cytometry, in homogenates, or IHC in histological sections [13]. In addition, the CAM assay has been used to perform drug efficacy studies, with targeted drug delivery to aggressive thyroid cancer cells of CAM-xenografted tumors (see paragraphs below). The proliferation, apoptotic/necrotic, and invasion phenotypes are detectable by both H&E staining and IHC for tissue specific markers and cleaved caspase 3 [14]. A clear readout of the tested treatment on cell death is achieved [15]. The CAM can also support patient-derived xenograft (PDX) development. Here it preserves the tumor heterogeneity and characteristics of the patient´s tumor, thus serving as a good platform for precision medicine [9]. More information is given under 8.3.

### 3.3. How Can the CAM Model Be Used to Study Cancer Cell Stemness?

Cancer stem cells (CSCs) or tumor-initiating cells (TICs), are usually a small subpopulation of cells with the ability of self-renewal, and this property has been considered a new hallmark of cancer [16]. Several types of CSCs may exist within the heterogenous tumor and their plasticity can be responsible for resistance to cancer therapeutics, frequently resulting in relapse [17,18]. Thus, a better characterization of CSCs may lead to alternative therapeutic targets able to tackle the survival advantage of CSC-enriched cancers. The CAM assay constitutes a valuable tool to functionally characterize cancer cells, thus also being a suitable model to study their stemness potential. When generating CSC-enriched clones from hepatocellular tumor cell lines by spheroid formation and single-cell cloning, the increased tumor-forming capacity in vivo with a more aggressive stem-like and invasive phenotype could be verified [19]. When CSCs were sorted by sedimentation field-flow fractionation (SdFFF) followed by 3D Matrigel amplification and grafting on CAM, the ovografts formed from CSC-enriched populations (CD44^+^, EpCAM^high^, CD166^+^) were highly invasive and vascularized [20]. When studying breast cancer brain-tropic cell lines in the CAM model, these metastasis-derived cells formed large and more compact tumors, with increased expression of CSC marker CD44 [21]. Given the importance of the tumor microenvironment as a modulator of cancer stemness, the gold-standard method to evaluate CSCs self-renewal ability are in vivo limiting dilution assays (LDA). A detailed chicken CAM-LDA has been described for a panel of breast cancer cell lines using tumor histological analysis and immunohistochemistry (IHC) staining for stemness markers. In comparison to mice LDA and mice tumors, CAM-LDA allows a more robust statistical analysis of CSC frequency and the histological analysis revealed that CAM tumors faithfully recapitulate human tumors [22]. Thus, the CAM model is also promising for testing the efficacy of potential anti-CSC drugs as shown for a novel hybrid compound based on 5-fluorouracil and thymoquinone [14]. Recently, Pizon et al. transplanted primary spheroid cultures derived from circulating CSCs of breast cancer onto the CAM. The CAM patient-derived xenografts resembled the primary tumor in histology, aggressiveness, and proliferation capacity [23].

## 4. Tumor Metastasis in the CAM Model

### 4.1. Is the CAM a Complete In Vivo Model for Studying Metastasis?

Invasion and metastasis are among the most prominent hallmarks of cancer [24] and the in vivo CAM system is highly useful in their study. Most cancer cells seeded on (or into) the chorioallantoic membrane (CAM) are, depending on their phenotype and aggressiveness, able to form a primary tumor. In some cases, provision of a 3D structure during implantation (i.e., Matrigel^®^) or slight tissue damage to the CAM surface through mechanical stress is needed for successful engraftment, while other cell lines will readily form a primary tumor when seeded as single cells in a saline solution (i.e., PBS). Following the acquisition of the surrounding vasculature and the induction of neoangiogenesis, most CAM tumors form expanding primary tumor nodules, comparable to other in vivo models.

Due to the complex and dynamic microenvironment of the in vivo CAM, the cells of the primary tumor have to overcome multiple barriers to infiltrate the surrounding tissue. Once the cells have completed the metastatic cascade (Figure 3), cancer cells from the primary tumor site will disseminate throughout the embryonic and extraembryonic tissue.

If these single circulating tumor cells (CTC) or tumor cell clusters reach different parts of the CAM or the organs of the avian embryo, they may be able to extravasate and are referred to as disseminated cancer cells (DTC). Only if they survive and are able to induce proliferation in this second, complex, and dynamic microenvironment of the metastatic site, may they be able to form micro-, and later, macrometastases [25,26].

As the completion of these complex and highly inefficient steps of the metastatic cascade takes time, it may be necessary to start such experiments earlier than others in the development of the CAM (i.e., on ED day 5–7) or to continue the experiment until ED 19 (bearing in mind the developing adaptive immune system and the gestation age of the embryo). Using these approaches, some aggressive cell lines are even able to produce micro- or macrometastases during the time on the CAM.

### 4.2. How Can the CAM Model Be Used to Study Tumor Invasion at the Primary Site?

The CAM is rich in vasculature and extracellular matrix proteins, such as fibronectin, laminin, type I collagen, MMP-2, and integrins, providing a physiological environment to study tumor cell intrinsic invasive properties. CAM assays have been used to study primary tumor invasion of numerous cancer entities, including colorectal [27,28], prostate [28], brain [12,28], gastric [29], thyroid [30,31], ovarian [32], lung [28], liver [33], head and neck [34], and breast [35] cancer. Based on the in vivo behavior, invasive scoring systems using tissue sections of the invading CAM tumor can be defined, providing a semi-quantitative method to evaluate CAM local invasion with significate detail. For example, Palaniappan et al. developed CAM-Delam, a standardized method defining different levels of the ability of human cancer cells to degrade the basement membrane (delamination) [28], an essential step in the invasion of surrounding tissue. The CAM assay can also be used to evaluate the effect of differential expression of genes of interest on the invasive ability of tumor cells (e.g., for gain- or loss-of-function studies) [29,36].

Additionally, the CAM model can be used to study the effects of different tumor microenvironments on tumor cells [26]. It has been shown that hypoxia generates a more invasive phenotype of SW480 colon carcinoma cells [37] and promotes ovarian cancer cell invasion via Snail-mediated upregulation of MMP1 [38]. Other mechanisms involving local cell invasion, such as autophagy, can also be studied in detail using the CAM assay. Jansen et al. provided a CAM-based method to study autophagy in esophageal adenocarcinoma tumors [39], showing that OE19 tumor cells show invasive growth into the CAM stroma, with a globular pattern and histology comparable to mouse xenograft tumors [40]. In these xenograft studies, tumor cells can be detected using human-specific antigens [39]. Moreover, peritumoral budding as an invasive marker defined as single cells or small cell clusters up to four cells can be quantified by pan-cytokeratin staining and evaluation of the tumor invasion front [36].

### 4.3. Can the CAM Model Be Useful in Tumor Metastasis Research?

Due to its ease of use, low bureaucratic requirements, complete in vivo biology, and low associated costs, the CAM model is ideally suited for metastasis research, which often requires high numbers of biological replicates. As defining for all in vivo systems, the complete metastatic cascade (Figure 3) can be studied in the CAM, even though some limitations that are due to time restraints apply. The use and versatility of the CAM assay in local tumor infiltration studies have been described above.

As a complete in vivo model, the CAM also allows examination of the metastatic abilities of xenografted human tumor cells. Once tumor cells leaving the primary tumor gain access to the CAM vasculature, the cells may spread hematogenously through this biological system. While there is scant research, the CAM also provides lymphatic vessels and a lymphatic spread is biologically possible. As cells invade blood vessels, CTC and disseminated tumor cells may be detectable in transit in the blood or lodged in secondary sites of the CAM and the developing embryo. While the short time-frame of the CAM assay (7–10 days) does usually not allow for macro-metastasis formation, cell clusters can, depending on the phenotype of xenografted cells, be detectable in lung, liver, or bone of late stage embryos [25]. The timeline of the embryonic days and the typical experimental window is excellently summarized in [41].

One frequently applied experiment to test the metastatic capabilities of xenografted cells is the so-called “lower CAM assay”. In this assay, following primary tumor formation on the “dropped” CAM, a specified amount of CAM tissue (i.e., 5 × 5 mm) perpendicularly opposed to the implantation site is harvested after opening the egg. The thin CAM tissue is then screened for extravasated human tumor cells. In most cases, the cells will be tagged with some form of fluorophore (i.e., GFP or cell dye), and thus DTCs can be detected and quantified by fluorescence microscopy either in vivo or, following enzymatic digestion of the tissue, in the resulting cell solution [26,42]. Thoroughly flushing the vasculature of the dissected CAM tissue reduces the amount of blood and, thus, CTCs in the specimen. Another approach to detect DTC is the use of (human)-specific PCR to test for the presence of human transcripts in the digested lower CAM tissue. This approach, if thoroughly controlled, can also be used to quantify the amount of disseminated cells [43].

The detection of metastatic cells in the highly replicative CAM system can also be useful in comparing the metastatic capabilities of different cell lines. Here, human-specific Alu sequences, repetitive sequences interspersed over all chromosomes, are used to detect cells of human origin. Using human-specific Alu quantitative real-time PCR (qPCR) of lower CAM and embryonic lung tissue, Zijlstra et al. [43] could demonstrate distinct, rate-limiting steps in the metastatic process of two different human cancer cell lines (HNSCC and fibrosarcoma). The Alu qPCR method has been further improved to reach a high sensitivity [44].

Furthermore, the effect of pharmacological treatment of primary CAM tumors on the amount of microscopically detectable metastatic cells (quantification) or their biological behavior and interaction with the host microenvironment in different organs of the developing avian embryo (i.e. bone, lungs or liver) may shed light on the (anti-) metastatic capacities of different compounds [25,26]. Such pharmacological testing is covered in detail in chapters 7 and 8 of this issue.

## 5. Angiogenesis on the CAM

### 5.1. Why the CAM Assay May Be Considered a Suitable Experimental Model to Study Tumor Angiogenesis?

All studies of mammalian neoplasms in the CAM have utilized solid tumors and cell suspensions derived from solid tumors. Grafting of tumors onto the CAM allows the study of the morphological aspects of the interactions of the tumors with the blood vessels of the host and the examination of the identity of the vessels that supply the grafts. The formation of peripheral anastomoses between host and pre-existing donor vessels is the main and the most common mechanism involved in the revascularization of the graft of an embryonic organ onto the CAM, whereas sprouting of CAM-derived vessels into the transplants only occurs in the grafts of tumor tissue [45,46].

When tumor cells are applied onto the CAM dependent on their aggressiveness, they attract CAM vessels to grow into the tumor ovograft to supply it with growth factors. Vessels that are seen in the ovograft have to be of embryonic origin. Thus, the CAM ovograft cultivation resembles a bioreactor with blood supply. Compared with mammals’ models, where tumor growth often takes between 3 and 6 weeks, CAM assay is faster. Between 2 and 5 days after tumor cell inoculation, the tumor xenografts become visible and are supplied with vessels of CAM origin [47]. Tumor cells can be identified in the CAM, as well as in the internal organs of the embryo such as lungs, liver, and brain. The latter is only possible when tumor cells have the capability to intravasate into the embryonic vessels.

Hagedorn et al. [48] confirmed this statement. They developed a human glioblastoma multiforme experimental model in the CAM and demonstrated that human glioblastoma multiforme cells implanted on the CAM formed avascular tumors within 2 days, which progressed through vascular endothelial growth factor receptor-2 (VEGFR-2)-dependent angiogenesis. Blocking of VEGFR-2 and platelet-derived growth-factor receptor (PDGFR) signaling pathways with small-molecule receptor tyrosine kinase inhibitors blocked tumor growth. Moreover, gene regulation analysis during the angiogenic switch by oligonucleotide microarrays identified genes associated with tumor vascularization and growth.

Recently, viral nanoparticles have been used to visualize newly formed vasculature in expanding tumors [49], and high-resolution imaging of CAM-supported human tumors reveal fluid and small molecule dynamics within tumors [50].

### 5.2. Can Imaging Be Applied to Monitor Angiogenesis and Tumor Growth?

Imaging techniques are of great importance in assessing and quantifying tumor growth and/or angiogenesis. There is a large body of literature describing the use of different imaging techniques for tumor growth and angiogenesis monitoring for different applications. In direct comparison to tumor analysis by conventional histology, applying in ovo imaging enables longitudinal assessment of tumor characteristics and development in the same sample. Imaging techniques range from straightforward optical microscopy or even just digital photographs [51], over enhanced optical methods including bioluminescence [52], fluorescence [53], confocal, and multi-photon imaging, and photoacoustic approaches [54], to more clinically established imaging methods like ultrasonography [55], computer tomography (CT, [56]), magnetic resonance imaging (MRI, [57]), nuclear imaging methods (PET, SPECT, [58]), and combinations thereof to benefit from the specific strength of the different modalities ([59,60]).

The above-mentioned imaging modalities have been applied to evaluate angiogenesis and arteriogenesis [61] in preclinical research and partly been transferred to the in ovo application. Due to the underlying physical principles, the different imaging techniques provide different aspects of the required information. Due to the usual superficially located vascular structure of interest in CAM research, optical methods are often applied. Direct in ovo visualization of macrovessels can, e.g., be achieved using a simple digital microscope [62], often in combination with Doppler techniques [63], where microvessel assessment demands, e.g., confocal microscopy after fixation of the tissue under examination [64]. Of paramount importance is the quantification of the angiogenesis [65,66], especially when applied, e.g., for the assessment of drug efficiency. Here, deep-learning-based approaches are on its verge [67]. Even though providing outstanding spatial resolution, the application of direct optical imaging is intrinsically limited by the penetration depth and restricted to the analysis of superficial processes. Considering that, in the meanwhile, almost all imaging techniques have been translated to in ovo application, further options exist for assessing angiogenesis and related perfusion/blood flow. However, due to their limited spatial resolution, direct visualization of the vasculature is often difficult and special tracers (contrast agents) are required. Those include special compounds for perfusion/blood flow as, e.g., Gd-DTPA for MRI, radiopaque compounds in CT, or ^13^N-ammonia for PET, as well as dedicated tracers binding to specific molecular targets related to angiogenesis. An excellent review of the state-of-the-art in evaluation of angiogenesis and tissue vascularization is provided in the respective AHA scientific statement [61].

Similar to imaging of angiogenesis, the imaging of tumor growth has been shown by different imaging modalities. Optical microscopy [51] is straightforward but does not provide any three-dimensional details of the tumor. Fluorescence [53] and bioluminescence [52] imaging have been proposed for longitudinal monitoring of tumor growth on the CAM. For (quantitative) assessment of the tumor growth, modified cell lines are required for providing the respective fluorescence or bioluminescence signal. For quantitative assessment of the 3D structure/volume of the tumor ultrasonography [55,68], MRI [57,69], and CT [70] with MRI providing more detailed characterization of the tumor because of its superior soft tissue contrast [57] has been applied. In combination with highly sensitive molecular imaging modalities like PET, tumor growth quantification can be combined with the quantification of metabolic and molecular assessment of the tumor [71].

From the current perspective longitudinal assessment of angiogenesis and tumor growth by in ovo imaging appears feasible. However, the respective imaging methods applied in a certain study need careful consideration to ensure accurate assessment of the envisaged information.

## 6. The Chicken Immune System

### 6.1. Is the Immunological Situation of the Eggs Comparable to the Human Situation?

The question of species dependency and comparability of immune reactions in developing eggs with the human situation plays an important role in many aspects of drug development, such as efficiency and toxicology testing of drugs and their drug delivery systems, tumor transplantation, metastasis, angiogenesis, and treatment options for autoimmune diseases, as well as the investigation of new immune-based drug principles [72,73]. All these investigations critically depend on a thorough understanding of the development and functioning of the immune system, as well as the role of age- and species-related variations in the quality and magnitude of immune responses. Moreover, the availability of reagents specific to the chicken species, like cytokines or antibodies for immunology-related lab experiments, can sometimes be challenging.

A paucity of information about the avian immune system in the literature is focused on hatched and mature organisms and is mainly derived from the poultry industry’s interests [74,75]. The situation for the in ovo and ex ovo embryonic egg test models is more complicated as they are used at different stages of embryonic development. Indeed, the developing embryo is characterized by a natural immunodeficiency that is gradually built up in three stages [72,76,77]. The immune system starts to be functional after embryonic development day (ED) 10–12, followed by a second phase up to ED 15 when immune responses comparable to the mature host start to develop. In the third stage, from ED 16 to hatching, the embryo develops resistance to most pathogens for the adult stage. Additionally, CAM experiments extending ED 15 were limited by nonspecific inflammatory reactions [78]. Under consideration of the development-related differences, the late-stage chicken embryo is a valuable model characterized by a robust but still partially sufficient immune system for immunologically relevant studies. Although the embryonic immune system is simpler than in humans, it is functionally similar and capable of building comparable immune responses [73].

### 6.2. Does Tumor-Induced Inflammation in the CAM Model Parallel the Typical Immune Cell Infiltration Observed in Human Tumors?

It is difficult to accurately determine how much tumor-induced inflammations in the CAM model parallel those in humans. Indeed, the inflammation process is particularly complex, and even though the chicken embryo’s immune system is relatively well-known, there are still many further explorations that are yet to be done. Moreover, even in humans, cancer-related inflammation is not fully known and is still under investigation [79]. Some ongoing studies aim to properly analyze the inflammatory capabilities of the chicken embryo, and the properties of tumor-induced inflammation in this model are being explored as well.

In humans, tumor-induced inflammation is described as a disturbance within the tumor microenvironment (TME), an intricate cellular ecosystem where immune cells, stromal cells, and tumor cells coexist and interact with each other [80]. While chronic inflammation is known to be potentially involved in carcinogenesis, with the exhaustion of immune cells and the acceleration of cell proliferation and angiogenesis [79,81], it has also been reported that tumor cells themselves can disturb the functions of the immune cells. Indeed, studies have shown that cytokines produced within the TME can promote cancer development, metastasis, and drug resistance [82]. Furthermore, immune cells recruited inside the TME, such as neutrophils and macrophages, can possess altered phenotypes and promote disease progression [83,84].

Even though there is a lack of knowledge concerning tumor-induced inflammation in the CAM model, evidence still points out that chicken embryos are capable of inflammatory reactions. Indeed, heterophils, the functional homolog of the mammalian neutrophils, can be recruited to an inflammatory site stimulated with lipopolysaccharide (LPS) as early as ED 7 [85]. While most studies on heterophil functionality occurred when the egg has already hatched, these cells appear to be already functional by ED 18 [86,87]. Furthermore, stimulation of the chicken embryo with Toll-like receptor (TLR) ligands can upregulate the expression of pro-inflammatory cytokines within the model [88,89]. Even though most cytokines are produced at higher levels after hatch, the embryo can still produce detectable levels of interleukin (IL)-1β, IL-4, IL-6, IL-8, IL-10, and interferon gamma (IFN-γ) [73]. All these cytokines have been reported to be involved in tumor-induced inflammation in humans [82]. Finally, it has been found that several types of chicken immune cells, namely the cytotoxic T cells (CD3^+^ CD8^+^ cells), helper T cells (CD3^+^ CD4^+^ cells), leukocytes (CD45^+^ cells), and NK cells (CD56^+^ cells), can be identified within the TME of a human xenograft in ovo, demonstrating the immune cell infiltrations from the host inside the tumor [90,91]. All this evidence shows that tumor-induced inflammations in the CAM model might be close to those observed in humans, even though it should be further investigated.

### 6.3. How Does the Developing Chicken Immune System Respond/Contribute to Human Tumor Xenografts?

The timeline of the CAM model’s embryonic development is key for assessing the contribution of its immune system. Indeed, the embryo gradually gains its immune competency over time, from a mostly immunodeficient model at ED 9 to one with a fully competent immune system at ED 18 [73]. This creates an ideal environment for the xenograft, as it allows the human tumor cells to be transplanted onto the CAM with a low risk of tumor rejection [92]. However, as the model’s immune system gets increasingly complex, the existence of immune cells has been confirmed in the TME of the xenograft. Among them, various T cells, NK cells, and other leukocytes have been identified [90,91].

However, to this day, it is not well-known at which exact point the immune system becomes complex enough for xenograft targeting. Studies have nevertheless demonstrated that early immune responses against the xenograft only occur after ED 12, with a visible infiltration of avian heterophils within the TME [93]. Furthermore, while conditions for xenograft rejection have not been properly established yet in the model, it is hypothesized that this would likely happen after ED 18 when the immune system is fully mature [94,95]. Since the use of the CAM assay usually ends at ED 18, there is, therefore, minimal risk of xenograft rejection during assays on the CAM model. This allows the chicken embryo to be considered an interesting model for studying immune cell infiltration.

## 7. The CAM Model in Immuno-Oncology

### 7.1. Can the CAM Model Be Used as a Powerful Model for Immuno-Oncology?

With the clinical application and success of novel immunotherapies, such as checkpoint inhibitors, CAR-T therapy, and various cancer vaccines, immuno-oncology (IO) represents one of the most promising areas of cancer therapy. To go farther on this road, relevant IO platforms, including pre-clinical in vivo IO models capable of mimicking the patient conditions, are indispensable.

To date, murine models are the most-used animal models in IO research, which are majorly represented by either human xenografts in immuno-compromised mice, or by mouse tumor xenografts in immuno-competent syngeneic mice [96]. Each model shows advantages, while the disadvantages are also obvious, such as the immunodeficiency nature of the former that renders it generally inadequate for many IO investigations and the latter being limited by only being able to test surrogate molecules that target the mouse immune system/tumors. Otherwise, genetically engineered mouse models and humanized mouse models have been recently developed, whereas the potential genetic alterations, the graft-versus-host disease, and the bad prediction of toxicity could also be the criticisms [97].

To answer these concerns, the CAM model is becoming a promising model for IO testing because this is a non-genetically modified, immune-competent in vivo model that permits the study of the interaction between human tumors and the model’s immune system. In fact, there exists an evident similarity in terms of the structure and functions of immune systems between human and chicken [73]. During the embryonic period, a chicken embryo gradually develops its immune system and increasingly builds its complex immune competency over time [75,78,93]. Due to this gradual development, the immune functionality in the chicken embryo before ED 9 is partial and limited, sometimes still considered immune-deficient at least before ED 10 [92], which indeed allows the model to be grafted with human cells with a low risk of transplant rejection. After that, with the development and maturation of different immune components, the immune responses occurring in the chicken embryos could be sufficient for immune investigations. Actually, different functional lymphoid cell populations, such as T cells, B cells, NK cells, antigen-presenting cells, and granulocytes, can be detected during the second half of embryonic development [73]. Furthermore, as many current immuno-oncology studies focus on immune checkpoint blockade, a considerable number of immune checkpoints targeted in current marketed immune-oncology therapies, including PD-1, PD-L1, CTLA-4, and LAG-3, have also been identified in chickens with great similarity to humans’ [73].

All these observations support the potential use of the chicken embryo as an in vivo model for predicting the efficacy IO drugs in humans. In conclusion, the CAM assay can be considered as an alternative, powerful, low-cost in vivo model for IO research allowing the screening of a large number of pharmacological samples in a short time [98].

### 7.2. How Can the Model Be Used for Drug Discovery and Testing?

During the drug development process, animal testing is currently preceding first-in-human studies. Preclinical animal models are used to investigate the toxicity, stability, pharmacokinetics, and mechanism of action of new bioactive molecules. Success rates in clinical trials are as low as 3% in oncology or 15% in neurology and question the current methodology of assessing investigational new drugs in preclinical science [99].

The major bottleneck of preclinical research resides in the complex architecture of organs like the brain or the immune system, as well as the multifactorial etiology of diseases. Despite the relevant reduction in drug candidates in early and advanced preclinical testing, too many substances still pass the preclinical phase but fail in clinical studies. Rethinking preclinical drug development will avoid expendable applications to human and animal test subjects and should reduce costs.

The CAM provides a technically simple way of studying complex biological systems with well-developed vascular tissues. It also has a high reproducibility; it is inexpensive and easy to handle. Therefore, the CAM model could be a model bridging the gap between cell-based and animal-based assays [32,85] (Figure 1).

In terms of molecules, the CAM model is suitable for a large panel of agents from small-molecule drugs to large biological compounds like antibodies, complex particles (nanoparticles, proteoliposomes, or viruses), or drug-delivery systems [100]. The major limitation is the solubility of molecules. In the mouse, non-soluble compounds are usually given in food or water. In the CAM model, this is not possible, and molecules must be solubilized to be injected. For local applications, excipients like olive oil can be used, for systemic injection, low-concentrated organic solvents like DMSO or ethanol. By chance, many vehicles like micelles, nanoparticles, etc. can be used, which generally permit the bypassing of this limitation.

In biomedical research, the CAM has been widely used as a model for angiogenesis studies because of its thin, transparent, and vascularized structure [101] and in xenograft tumor studies for invasive human tumor cells because of its immature immune system [85,102]. Moreover, the CAM has been accepted as a substitute for the Draize test on rabbits for the testing of the irritation potential of chemicals [103,104]. The conventional CAM assay administers drugs through implantation of a membrane or coverslip containing the investigated compounds to study neovascularization and/or to study the injury on the CAM. During the development of drug-delivery systems, chicken embryos can be used to evaluate the activity or toxicity of a drug in both the CAM and CAM-grafted tumors, as well as on the development of the embryo. Toxicity of drugs or carriers on chicken embryos can be evaluated in terms of embryo death or adverse effects on the CAM, including inflammation and neovascularization.

### 7.3. How Widely Can the CAM Assay Be Used for Oncology Drug Discovery?

As a real in vivo tool, the CAM assay emerges as a useful, reliable, complementary pre-clinical model for oncology drug discovery. First, this model is suited for testing a large range of anti-cancer therapeutics. Indeed, the CAM assay has already been reported in pre-clinical screening to assess the efficacy of various anti-cancer agents, such as small molecules [105], oligonucleotide [106], peptides [107], antibodies [108], proteins [109], virus [110], natural substances [111], etc., against different types of cancer. Because the chicken egg is a self-sustained and closed system and, therefore, there is no excretion of the substance [112], additional benefits of using the CAM model, such as the lower drug-quantity requirement and the reduced administration frequency, could be interesting for drug screening. Otherwise, if drugs, and especially small molecules, are not metabolized, they could re-enter the embryo and possibly increase the toxicity of drugs. Thus, after the validation of compound efficacy, the adequate dose for transfer into human applications must be redefined. In addition, the CAM model can be used to test the agents developed for different cancer therapies, like chemotherapy [113], radiotherapy [13], photodynamic therapy [114], and nutritional therapy [70], in monotherapy, as well as in a combination setting [115,116].

Besides, the growth of tumor cells grafted on the CAM can faithfully recapitulate most of the characteristics of the carcinogenic process [47], which is therefore a multifaceted experimental model allowing the study of the anti-tumor performance of drugs in different aspects, such as in vivo toxicity [117], tumor growth [12,118], metastasis invasion [32,113], angiogenesis [119,120], metabolomics [121], transcriptomics, etc.

Moreover, the CAM assay is also a suitable model to evaluate drug-delivery systems for cancer treatment. Actually, drugs can be administrated in ovo in different ways, either by topical application, intra-tumoral injection, or i.v. injection. Therefore, different biological barriers can be examined within a single model [122]. In addition, to overcome the challenges imposed by conventional treatments, like the limited therapeutic efficacy and the evident side effects, the development of drug-delivery systems with novel technologies, such as nanotechnology, demonstrates a promising potential [100]. For example, Vu et al. demonstrated that the loading of doxorubicin with periodic mesoporous organosilica nanoparticles led to an excellent tumor accumulation of the nanoparticles and, therefore, to the specific delivery of doxorubicin to a human OVCAR-8-initiated tumor, which resulted in the effective in ovo elimination of the tumor and particularly in reduced organ damage [123].

It is also possible to check the CAM model pharmacokinetic (PK) [57] and pharmacodynamic (PD) [124] parameters of drugs [125], which are vital in drug discovery. With the recently reported applicability of the CAM model for blocking studies for evaluation of receptor-specific radio ligand binding, its applicability for the evaluation of binding specificity has also been shown [126]. In summary, the CAM is a relatively simple and reliable model that allows the screening of many anti-cancer agents in a short time.

### 7.4. Can Imaging Be Applied for Initial Testing of Pharmacokinetics and Bio-Distribution?

The pharmacokinetics or bio-distribution of a substance describes the process of absorption, distribution, metabolism, and excretion in the corresponding biological system. Prerequisites for a successful therapeutic substance are the stability of the substance, its access to the target region in the system with simultaneous low off-target accumulation, and a sufficient retention time in the target region to achieve the therapeutic effect. In addition, there is the degradation or excretion after the drug has taken effect at the target site [127].

The chicken embryo model has several disadvantages with regard to pharmacokinetics, which must be taken into account and studied in detail for the specific analyses. The chicken embryo is still in the developmental stage with most organs, some of which even gain some of their functions after the chicken hatches, e.g., brain and lungs [128,129,130,131]. The organs of the reticuloendothelial system, particularly the liver and kidneys, must be as fully functional as possible for pharmacokinetic analyses. This limits the time window for these studies to a maximum of the last third of the total developmental period of the embryo, i.e., beginning at the earliest with ED 15 [112,132,133,134].

In addition, the distribution of a newly developed drug is mostly of interest with respect to its future use in humans. Consequently, regardless of the developmental stage of the organ, similar or dissimilar morphology in the model organism is also important [135]. For example, avian kidneys have structures from both mammalian and reptilian kidneys [132,136,137], but differently, the chicken excretes the nitrogenous waste as uric acid; in mammals, urea is excreted. Since a urinary bladder is lacking in the chicken, there is no urine storage capacity [138]. Compared to body size, the avian liver is larger than those in mammals; there is less connective tissue and no true lobular structure [139,140]. Although the metabolic processes in the liver of birds and mammals are similar, there are differences, for example, in lipogenesis [139,141]. Regarding excretion and resorption, the gut system of birds should also be considered for pharmacokinetics. Compared to the mammalian stomach, posteriorly to the proventriculus a second sacular, disc shaped and very muscular structure, the ventriculus or avian gizzard, is formed starting at ED 6 [142,143]. The gizzard´s expression of proteins or receptors may have an influence on the pharmacokinetics of the studied drug [144,145].

In addition to anatomical differences, metabolic differences between birds and mammals, depending on the substance under investigation, should be considered for the evaluation of pharmacokinetics [146].

Although there are a number of factors that need to be considered in pharmacokinetic analyses in the chicken embryo model, several promising publications already indicate that such analyses are possible. Pharmacological analyses have been performed in chicken embryos for many years, as teratological analyses provide indirect information on the pharmacokinetics of a substance. The combination of teratological and pharmacokinetic studies has allowed more accurate results and the possibility of cross-species predictions [147,148,149,150,151,152,153].

Furthermore, toxicological studies, often closely related to teratological studies, also provide evidence for pharmacokinetic effects of compounds and have been repeatedly performed in the chicken embryo model [15,117,154,155,156,157]. While teratology and toxicology studies usually represent end-point analyses that interpret the distribution of the compound based on the final effect, time-resolved studies allow for actual pharmacokinetic analyses.

Analysis of metabolites of applied compounds in the chicken embryo brain at various time points provides an important methodological basis for pharmacokinetic studies in this in vivo model [158]. The uptake kinetics of adriamycin in the heart and liver of chicken embryos were published by Johnson et al. in 1986 [159]. In addition, the study of uptake kinetics in tumors by Hornung et al. 1999 is evidence of the feasibility of pharmacokinetic studies in the chicken embryo model [160]. The analysis of the accumulation or the effect of the drug on various off-target organs of the chicken embryo represents pharmacokinetic studies in the true sense [161,162]. The pharmacokinetics of acetyl-boswellic acids in the plasma of chicken embryos was studied by Syrovets et al. in 2005 [163], while Vargas et al. specifically analyzed the chicken embryo model for the possibility of pharmacokinetic studies in 2007 [164]. Recently, pharmacokinetic analyses based on imaging techniques have been published, which are highly promising for the analysis of new compounds with respect to off-target binding [60].

Even pharmacokinetic modeling was successfully performed by Yadav et al. based on chicken embryo experiments [165]. This study clearly demonstrates the potential of the chicken embryo model for pharmacokinetic analyses. The work published to date demonstrates the great potential of the chicken embryo model for use in pharmacokinetic studies of newly developed compounds. The limitations of the model must always be critically considered.

## 8. The CAM Model in Personalized Medicine

### 8.1. In What Ways Will the CAM Model Facilitate and Extend Possibilities for Developing Personalized/Precision Medicine Solutions in Oncology?

Advances in pharmacogenomics have dramatically increased the potential for customizing health care for individual cancer patients. This has been made possible through the complete sequencing of the human genome and better characterization of the human epigenome, proteome, and metabolome. Personalized medicine has made systematic use of a patient’s genetic profile and biomarkers to select or optimize therapeutic care. Molecular profiling in healthy and cancer patient samples may allow for a greater degree of personalized medicine than is currently available. Despite this progress, many cancer types are not yet addressable by these approaches. Furthermore, the heterogeneity of the tumor for individual patients remains an issue for this kind of analysis. 

Recently, a new avenue for personalized medicine in cancer has been developed, using patient-derived tumor models to directly test the efficacy of known treatments for each patient. This approach has the advantage of directly studying the tumor response empirically, thus taking into account the full heterogeneity of the individual tumor. This approach was first tested using the mouse model, but because of the poor engraftment rate and the long time required for tumor growth, the mouse model was determined to be incompatible with clinical use. Indeed, to be useful for patients, a recommendation for the best cure is needed within three weeks, along with large-scale applicability via a high successful engraftment rate for patients. 

The CAM model allows growing patient-derived tumors, for all types of tumors, with a high engraftment rate and a rapid analysis time that is compatible with clinical needs; it is a unique and promising alternative solution for personalized medicine. It allows a better characterization of the patient primary tumor’s heterogeneity at the cellular and genomic level, the capacity for metastatic invasion, and finally identification of the most efficient cure for each patient via a direct in vivo assay. The CAM model seems to be a very good choice with its rapid growth, low cost, reproducibility, robustness, and reliability in reproducing human tumors. This was demonstrated by the direct use of patient primary tumor biopsies on the CAM in many examples [95]. More recently, as successfully described by Pizon et al. [23] and Rousset et al. [166], the in ovo CTC-derived xenograft (in ovo CDX) could provide an avatar of the patient’s tumor. CTCs are heterogeneous and rare in the bloodstream but responsible for cancer metastasis. Overall, this new perspective could allow identification of the subpopulations leading to metastatic spreading and treatment resistance and, thus, cancer progression or recurrence. The challenge is to further explore these tumor phenotypes and, what is more challenging, their reliable expansion.

Thus, the CAM assay will allow the emergence of new individualized tumor models, representative of the patient’s tumor and CTCs’ heterogeneity. This provides access to drug screening while respecting a short timeframe to allow timely guidance for therapeutic decisions. These recent advances have launched significant new perspectives for the use of patient biopsies in personalized medicine.

### 8.2. In Particular, What Are the Specific Benefits for Personalized/Precision Medicine That Can Only Be Achieved Using the CAM Model?

The key advantages expected with the CAM assay as a routine therapeutic diagnosis tool or companion test would be: (A) it is extremely fast. First, the CAM model is the only one where the time between biopsy and starting a successful patient treatment is compatible with clinical requirements. Furthermore, the fast in ovo model allows fewer mutations and clonal selections, thus it more closely captures patient features by maintaining the heterogeneity and pathophysiology of the original tumor. (B) It is difficult to study immunologic drugs in the immunodeficient mouse strains used in most PDX models without first undergoing an elaborate and resource-intensive process of “humanization” to establish an immune system. In contrast, the chicken embryo model has a physiological immune system and is well-adapted to testing immuno-oncology drugs used in clinics, such as pembroluzimab [90], and allows testing IO resistance for individual patients. (C) The CAM model has been shown to be highly predictive, to assess efficacy as a chemosensitivity predication model for two established drugs in the treatment of gliomas, and a high degree of positive association was shown with the clinical outcome [167]. (D) Only low-profile facilities are required for the model. There is no need for animal facilities, which are labor-intensive and time-consuming and raise ethical issues. Chicken eggs could easily be managed in a hospital-associated laboratory with low additional equipment costs. Furthermore, from a regulatory point of view, as this is not considered to be animal testing, no prior ethical reviews are necessary. (E) This solution is highly affordable and compatible with health-insurance reimbursement levels because of the small lab resources needed. Furthermore, the cost of testing will be easily offset by the elimination of ineffective patient therapies.

Thus, we believe that the CAM model presents a great opportunity for personalized medicine by using a combination of genetic data and CAM experimentation to predict and test sensitivity of an individual patient’s cancer to specific treatments.

### 8.3. How to Leverage on the CAM Model as a Living Biobank to Facilitate the Precision and Personalized Medicine for Patients in the Real World?

To achieve precision and personalized medicine for patients in the real world, utilizing materials derived from the patients for diagnostic investigation or therapeutic testing is crucial. The aim is to find alterations such as gene mutations that could be actionable to be matched to specific therapeutic options. Having these patient-derived materials (PDMs) established in a living biobank setting could further facilitate more comprehensive investigation and testing in a sustainable way. One goal is the expansion of the PDMs to a larger quantity sufficient for downstream pipelines of molecular profiling and drug testing. Another goal is the preservation of the original characteristics of these PDMs to faithfully reflect the clinical status of patients. Moreover, the turnaround time of the molecular profiling or drug testing results would need to meet the timeline to make clinical decisions. Therefore, the model of choice to establish the PDMs should take these goals into consideration. Both in vitro and in vivo models have been adopted for this purpose. Among the in vivo models, the mouse model has been widely used to establish patient-derived xenografts (PDXs) by using tumor fragments derived from cancer patients [168] with good results for the first two goals. However, one bottleneck for the mouse PDX model would be the longer turnaround time, which could take up to months for the initial xenografts to be established. The CAM model, in this regard, is a great alternative solution since the average take time for PDX in the CAM would be in weeks [9]. The well-vascularized and naturally immune-suppressed CAM environment offers an efficient working model during the 3-week embryonic development window. The expansion of the PDXs could be achieved by serial passaging and subsequent grafting to new batches of fertilized eggs. The shorter timeline in the CAM model also means that the manipulation or the influences from the host system being introduced to the PDMs could be kept to a minimum.

### 8.4. Is the CAM Model an Animal Experiment and What Are the Consequences for Investigational New Drug (IND) Documentation?

From a regulatory point of view, the CAM assay is already recognized as an in vivo assay that is outside the scope of animal experimentation. If as is usually the case, the CAM assay stops 2 days before hatching, assays are not considered to be animal experimentation from a regulatory point of view in Europe (EU directive 2010/63/EU, [7]) and the USA. This implies that authorization for animal experimentation (e.g., IACUC in the USA) is not needed for the assays, but the American Veterinary Medical Association (AVMA) considers 72 h before hatching as a critical period [169]. Thus, all studies carried out with this model are true in vivo assays, without animal testing. The CAM assay is a true alternative method, which meets the 3Rs criteria, which are a huge challenge for pharmaceutical companies, corporate social responsibility (CSR), and a widely shared social issue. The 3R (replace, reduce, refine) objectives and alternative methods to animal experimentation are now highly recommended by regulatory bodies such as the FDA (CEDER), EMA, and ICH, and by the European Commission (2021/2784(RSP)). As the CAM assay is “zero animal use” (replace) and the in vivo data produced are predictive of results in rodents, this allows us to drastically reduce and refine rodent experimentation (Figure 4).

One critical point that has not been precisely determined is when exactly pain sensation is fully developed in the embryo, but nociception develops in chicken embryos similar to that in mammals [170] and the prerequisites for pain sensation are present from developmental day 7 [171]. Here, it needs to be considered that the CAM is not innervated and, thus, there is no sense of pain. Moreover, from a developmental biology point of view, the CAM is an extra-embryonic organ supporting the embryo as an early respiratory organ [4].

A huge challenge remains: the CAM model being included by regulatory agencies in their guidelines as an alternative method for the nonclinical testing of anti-cancer drugs. The CAM assay is already used in the study of angiogenesis inhibitors approved by the FDA [172] and is often mentioned in EMA and FDA reports [173]. Even so, the crucial inclusion of the CAM assay in the guidelines remains to be achieved. Nonetheless, CAM assay results are often included in the IND documentation and are well-accepted by the regulatory agencies. 

## 9. Limitation and Advantages of the CAM Model

### 9.1. What Are the Advantages?

Many advantages make the CAM an attractive model for cancer researchers (Figure 4).

First: the major advantages of the CAM model as an experimental model in oncobiology are as follows: the chicken embryo is naturally immunodeficient at early stages (before ED 10) and can accept cancer cells regardless of their origin without an immune response [73]; the changes in morphology of cancer cells arrested in the CAM microcirculation can be readily observed by in vivo microscopy; most cancer cells arrested in the CAM microcirculation survive without significant cell damage and complete extravasation; viral nanoparticles have been used to visualize newly formed vasculature in expanding tumors, and high-resolution imaging of CAM-supported human tumors reveal fluid and small molecule dynamics within tumors; and engraftment of human tumor tissue onto the CAM, followed by transcriptomic analyses with both human and chicken microarrays, enables the gene signatures of both the host stroma and the human tumor to be distinguished [93].

Second: despite a short observation period, the CAM tumors histologically resemble primary carcinomas and exhibit morphological characteristics of the original tumors, which is a huge advantage over tumor cells in culture.

Third: tumor spheroids and organoids can only show the features of real tumors to a limited extent as they are not formed in a rich tumor microenvironment. In this context, it is also of particular importance to consider the chicken embryo model because of the feasibility of live imaging, which can enable the tracking of tumor development, metastasis, and remodeling of the extracellular matrix [11,59,174,175].

Fourth: the cost of the basic equipment and the eggs themselves is very low. Therefore, even labs with a limited budget can readily set up a CAM lab, and since the CAM assay is acknowledged as a 3R (replacement, reduction, refinement) method, no special approval is required. Besides, although environment and experimenter could bias an in vivo experiment [176], it is easy to reduce variability for these parameters in CAM assays (e.g., no animal manipulation and exactly the same environment for all groups in the same incubator).

Fifth: studies in mouse models are done in standardized strains, without genetic heterogeneity between individuals (e.g., with the same genetic background). Therefore, researchers are now thinking of going back to less-standardized models to mimic heterogeneity in humans [177]. In the chicken embryo model, genetic background is not controlled, which could be an advantage compared to the mouse model. Furthermore, studies in this model using males and as well as females, with no specific selection, allows the mimicking of a mixed population. A last advantage is the rapid development of tumors. It takes only days to observe tumors of a few milligrams, while it takes weeks in the mouse.

### 9.2. What Are the Limitations?

First: the post-treatment observation period is short. Even if the rapid development of the tumor is an advantage for using this assay, the limited experimentation time restricts long-term observations. After xenograft at ED 9, there are only 9 days to treat and follow tumor development in ovo, until ED 18, to respect the regulatory rules about animal testing. So, the window for CAM experimentation lasts only 10 days. Even though this window allows the growth of a tumor in a few days in a rich embryonic microenvironment, we have only one week to test drugs and there is no possibility to study the remission, for example. Thus, it is not possible to observe the potential recurrence, but that can be realized in the mouse model.

Second: in terms of molecules, the major limitation is not related to the molecule classes, but sometimes to the solubility of the molecules. In mice, non-soluble compounds are usually given in food or water. In the CAM model, this method is not possible and molecules must be solubilized to be injected. Besides, the oral administration of drugs cannot be carried out in this model.

Third: in terms of readouts, although various analyses can be achieved using the CAM assay, some limitation exists. For example, it is difficult to achieve the blood collection before ED 13 because of the size of the embryo, and this collection is normally considered an end point for individual embryos because of its relative fragility; therefore, kinetics must be estimated at each time point in different eggs. Besides, because the chicken’s immune system has yet to be sufficiently clarified compared to the human’s and the mouse’s, the availability of immune reagents, like antibodies, for this model is less important than for humans and mice.

Fourth: monitoring of three-dimensional tumor size and growth might be the main methodological limitation of the CAM assay. Although a direct view of the CAM is possible after partial shell removal, the microscopical evaluation of tumor size and tumor vascularization is limited by the opacity of the surrounding eggshell, as well as by the autofluorescence of the solid tumor in fluorescence microscopy [174]. Generally, the tumor often grows deep into the CAM, resulting in a massive increase in tumor volume without major changes in lateral diameter. Furthermore, the movements of the chicken embryo further impede sufficient and reproducible microscopical investigation in ovo. An end-point analysis can be realized by tumor fixation and preparation of histological slides for immunohistochemical analysis of specific markers of tumor cells.

In addition, there are differences in terms of the drug’s metabolism and the immune system between chickens and mammals. Suchlike issues lead to the fact that drugs must be further examined in a rodent model. Hence, the egg can be considered as the appropriate model before using the mouse, fulfilling the 3R criteria.

### 9.3. When Debate Is Ending in a Tie, Is It However a Win-Win for Researchers?

In terms of cancer types, it is possible to graft sarcomas, carcinomas, and hematopoietic cancers as well. Komatsu et al. have reviewed 14 types of cancers in the literature [178] and Inovotion has validated 19 types of cancers for customer studies [179] until now.

Xenografts in ovo can be obtained from classical cancer cell lines [178], from patient biopsies [95], and even from CTCs [23]. All these models can be used to follow tumor development and metastasis invasion to the lower CAM and/or tissues in the embryo. The main limitation in terms of xenograft in ovo is, as for other in vivo models, the original quantity of available sample. This concerns tumor cells from patients and especially CTCs, which can be isolated in very small amounts from patient blood samples. To bypass this limitation, it should optimize tumor cell amplification in ovo to create PDX cell lines as has already been done in mouse models.

In terms of molecules, the CAM model is suitable for a large panel of agents from small-molecule drugs to large biological compounds like antibodies, complex particles (nanoparticles, proteolipsosomes, or viruses) or drug-delivery systems [100]. The major limitation is the solubility of molecules. In mouse, non-soluble compounds are usually given in food or water. In the CAM model, this manner is not possible and molecules must be solubilized to be injected. By chance, many vehicles can be used, which generally permits the bypassing of this limitation.

By the way, the chicken CAM model is a complete organism, allowing the use of biologics for efficacy and toxicity tests. If antibodies developed for humans require to be tested in ovo, this must be preceded by a sequence comparison to validate the binding ability of the antibody to chicken-origin receptor/antigen. For example, chicken and human possess a high cross-activity in the binding sites of pembrolizumab, an anti-PD1 antibody used in clinics, which allows the use of the CAM assay for pembrolizumab study [91].

In terms of readouts, there is indeed no difference between in ovo and other in vivo models. PK/PD measurement, toxicity estimation, efficacy evaluation (on tumor, metastasis, angiogenesis, immune cell infiltration, and imaging) can all be investigated in ovo, but some limitation exists in terms of sample/data collection from the same individual egg. Because of the size of the embryo and its relative fragility, it is difficult to collect blood before ED 13. In most cases, it is preferable to consider blood collection as an end point for an individual egg. Therefore, kinetics must be estimated at each time point in different eggs.

## 10. Conclusions/Outlook/Perspectives

In summary, the CAM model fulfils the 3R guidelines to replace and reduce animal experiments. Nevertheless, there is no general acceptance of the CAM model for different applications such as cancer research. Mouse experiments are still preferred in the scientific community. The major reasons seem to be, on one hand, the low level of standardization, particularly with a view to international harmonization and inter-laboratory comparability, and on the other hand, the lack of a direct comparison between mouse and CAM models for a specific application. Thus, the stated goals of the CAM community must be these two important topics and also the organization of the CAM researchers and regulators in a CAM society. Such a CAM society should collect and provide a database listing relevant applications for the CAM model in cancer research and affiliated branches that can be easily used as an information tool about alternative test systems.

## 11. Patents

PG https://patentscope.wipo.int/search/en/detail.jsf?docId=WO2020089561, (accessed on 2 November 2022) [91].

## Figures and Tables

**Figure 1 cancers-15-00191-f001:**
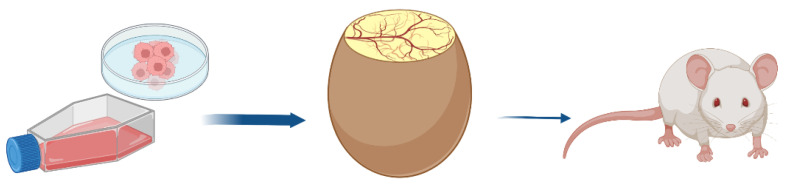
The egg before the mouse.

**Figure 2 cancers-15-00191-f002:**
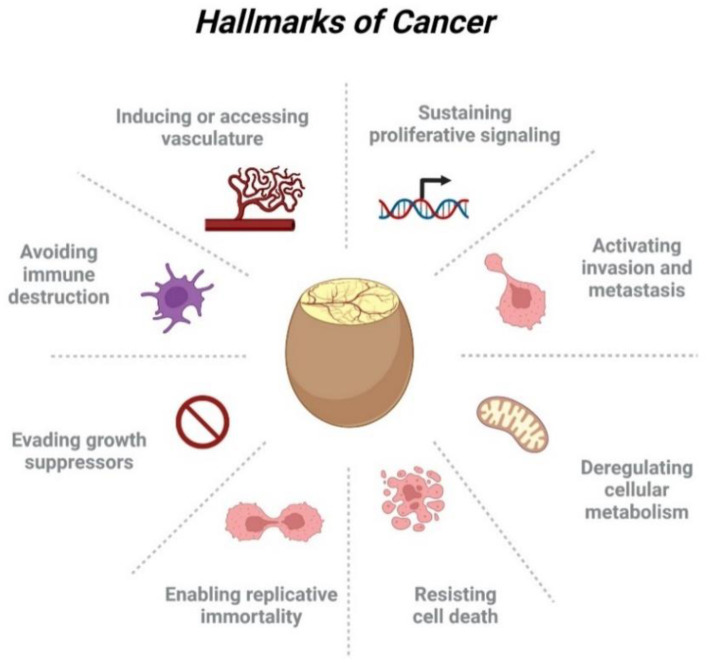
The CAM model and the eight core hallmarks of cancer.

**Figure 3 cancers-15-00191-f003:**
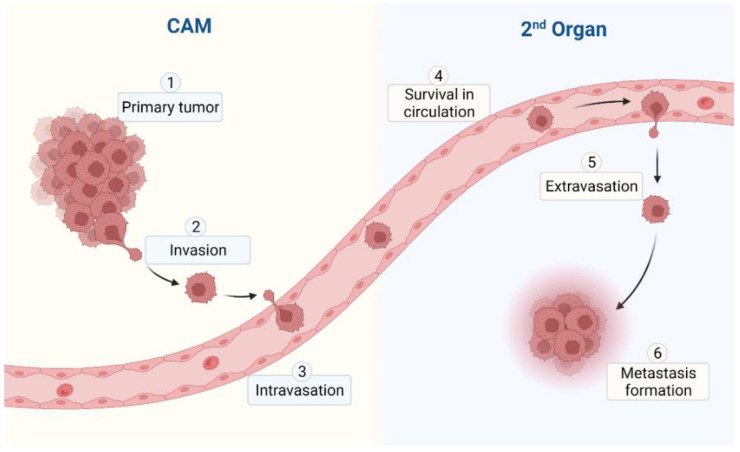
Metastatic cascade in the CAM model.

**Figure 4 cancers-15-00191-f004:**
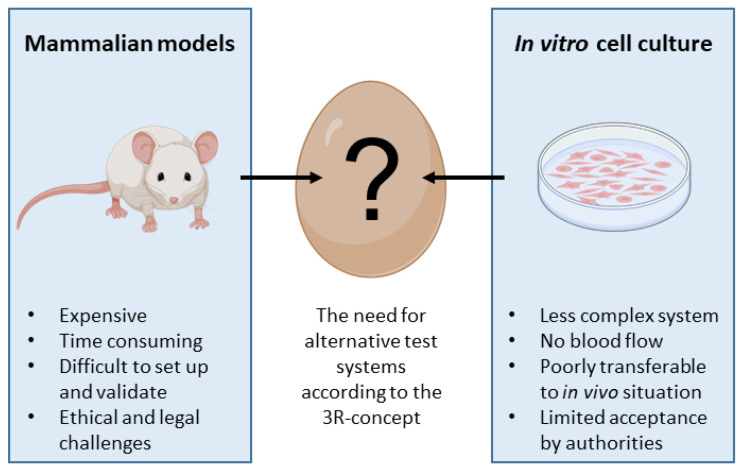
Scheme showing the intermediate position for the CAM model between mammalian models and in vitro cell culture.

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
