# Peer review of "The CAM Model—Q&A with Experts"

_cancers, 2022, doi:10.3390/cancers15010191_

Round 1
Reviewer 1 Report
The present review provides a very comprehensive overview of the use of the CAM in cancer research. I congratulate the authors fir this endeavour. The review is well written, however, some parts should be edited for English grammar. I have numerous, but mostly minor remarks that could enhance the manuscript further.
The structure of the article is good, although I would advise to put section two more towards the end of the article. Also a table of content would be usefull for this review.
Simple Summary:
The CAM model is more than just a model bridging in vitro and mouse research. PDX studies might also be directly relevant to choos patient treatments. Maybe consider rephrasing this.
Introduction:
Line 53: provide reference, there have been some papers in e.g. ALTEX describing the CAM as alternative model
Lines 71-77: it is confusing to go from ED14, back to ED11 and then to ED18
2.3. This section might be put later in the manuscript. maybe even the entire regularities- legal framework section should be considered to be moved towards the end of the manuscript.
3.1:There are more hallmarks of cancer than listed in the figure below. For example, enabling replicative immortality, genome instability & mutation, evading growth suppressors, ... Is the CAM model also suitable for these hallmarks?
Also on line 152 a reference is made to the hallmarks of cancer. However, the cited papers contain dated information. In 2022, a new paper was released by Hanahan D. (Douglas Hanahan; Hallmarks of Cancer: New Dimensions. Cancer Discov 1 January 2022; 12 (1): 31–46. https://doi.org/10.1158/2159-8290.CD-21-1059), which includes the currently accepted hallmarks of cancer. Hence, I would esuggest to edit the figure with more hallmarks, or mention that the figure does not include all currently accepted hallmarks of cancer. Or only mention in the figure the hallmarks for which the CAM model is (possibly) applicable. Elaborate why the CAM model is suitable for these hallmarks
Fig 1. this figure should be put near the text, although the authors might want to emphasize that bridging the gap is not the only role of the CAM model. It also serves an interesting role on itself.
lines 143-148: this paragraph seems a bit lost here. Maybe better to add the xenografting possibility to the paragraph before, and discuss the 3R principle etcetera earlier on, when figure 1 is first mentioned?
Line 157: the authors refer to cell lines, however, the CAM model can also be used as a PDX model for the growth of tumor biopts right from the patient. Maybe this option should also be included somehow, as the CAM system is especially suited for these kind of experiments.
179: this sentence can be deleted, since it is already explained
272-273: for readers who are not familiar with the CAM assay, it might be confusing why ther is only a 7-10 day time frame. Maybe a figure with the timeline of the embryonic days and the typical experimental window would help?
290: maybe provide a bit more background as to the reason why the alu-qRT-PCR is preferred.
Is the abbreviation qRT-PCR correct? RT stands for reverse transcription according to the MIQE guidelines: https://pubmed.ncbi.nlm.nih.gov/19246619/
Line 501: change "proofs" to "observations", and again a "the" missing before the "chicken embryo"
line 509: add a reference which established these percentages.
7.2: one problem with the CAM model is that it is a closed system. Although the liver and kidney do function. Most excreted products go to the allantoic fluid, which is partly taken up again. If drugs, and especially small molecules, are not metabolized, they could re-enter the embryo. This might increas the toxicity of drugs. I'm not sure if this has been investigated, but it should be discussed. it is written as a benefit in 551, but obviously also bears a drawback.
Lines 585-599: it should be discussed how similar these organs (kidney, liver and gut) function in comparison to mammals. At least the kideny will have some differences.
minor:
line 45 change interview to review
line 137: explain the abbreviation FACS
line 483: Change "to answer these concerns, chicken embryo..." to "to answer these concerns, the CAM model..."
Re-read part 7 for grammer, often, the article "the" is lacking where needed: such as in line 489: the immune functionality in the chicken embryo...
Author Response
Revision
Point-by-point answer to the comments of the reviewers
Reviewer 1:
The present review provides a very comprehensive overview of the use of the CAM in cancer research. I congratulate the authors fir this endeavour. The review is well written, however, some parts should be edited for English grammar. I have numerous, but mostly minor remarks that could enhance the manuscript further.
The structure of the article is good, although I would advise to put section two more towards the end of the article. Also a table of content would be usefull for this review.
Thanks for this valuable comment:
- We added a table of content summarizing all questions
-regarding section 2 see the answer below
Simple Summary:
The CAM model is more than just a model bridging in vitro and mouse research. PDX studies might also be directly relevant to choos patient treatments. Maybe consider rephrasing this.
- We rephrased the Simple Summary
Introduction:
Line 53: provide reference, there have been some papers in e.g. ALTEX describing the CAM as alternative model
-we added three representative references
Lines 71-77: it is confusing to go from ED14, back to ED11 and then to ED18
-we clarified this point
2.3. This section might be put later in the manuscript. maybe even the entire regularities- legal framework section should be considered to be moved towards the end of the manuscript.
-the CAM regularities are given now at the end of the manuscript, together with advantages and disadvantages,
-we prefer to show the general aspects of alternative anaimal testing and the SOPs at the beginning of the manuscript as a kind of introduction to the manuscript and starting point for the aspects of the following discussion
3.1:There are more hallmarks of cancer than listed in the figure below. For example, enabling replicative immortality, genome instability & mutation, evading growth suppressors, ... Is the CAM model also suitable for these hallmarks?
Also on line 152 a reference is made to the hallmarks of cancer. However, the cited papers contain dated information. In 2022, a new paper was released by Hanahan D. (Douglas Hanahan; Hallmarks of Cancer: New Dimensions. Cancer Discov 1 January 2022; 12 (1): 31–46. https://doi.org/10.1158/2159-8290.CD-21-1059), which includes the currently accepted hallmarks of cancer. Hence, I would esuggest to edit the figure with more hallmarks, or mention that the figure does not include all currently accepted hallmarks of cancer. Or only mention in the figure the hallmarks for which the CAM model is (possibly) applicable. Elaborate why the CAM model is suitable for these hallmarks
-we saved only the recent reference of Hanahan et al. 2022 as given already in reference 8 and omitted the earlier versions.
-we clarified that the hallmarks given in Figure 2 are the core hallmarks
-we elaborated more extensively why the CAM model is suitable especially for these hallmarks that are given in the Figure 2
-we re-wrote the point 3.1.
Fig 1. this figure should be put near the text, although the authors might want to emphasize that bridging the gap is not the only role of the CAM model. It also serves an interesting role on itself.
-this is an important point and we agree with the reviewer: we emphasized the extraordinary role of the CAM model itself
lines 143-148: this paragraph seems a bit lost here. Maybe better to add the xenografting possibility to the paragraph before, and discuss the 3R principle etcetera earlier on, when figure 1 is first mentioned?
-we re-structured this paragraph, changes are given in blue letters
Line 157: the authors refer to cell lines, however, the CAM model can also be used as a PDX model for the growth of tumor biopts right from the patient. Maybe this option should also be included somehow, as the CAM system is especially suited for these kind of experiments.
-we added: The CAM can also support patient-derived xenograft (PDX) development. Here it pre-serves the tumor heterogeneity and characteristics of the patient´s tumor thus serving as a good platform for precision medicine. DOI: 10.1159/000513039 More information is given under P.8.4.
-we gave a link to paragraph 8.4 about PDX application on the CAM
179: this sentence can be deleted, since it is already explained
-Thanks for carefully reading or manuscript: we deleted this sentence.
272-273: for readers who are not familiar with the CAM assay, it might be confusing why ther is only a 7-10 day time frame. Maybe a figure with the timeline of the embryonic days and the typical experimental window would help?
- we added a reference DOI: 10.1002/dvdy.24093 that excellently describes the time line of the CAM experiment
Is the abbreviation qRT-PCR correct? RT stands for reverse transcription according to the MIQE
guidelines: https://pubmed.ncbi.nlm.nih.gov/19246619/
Thanks for this comment:
-According to the MIQE guidelines we changed into qPCR (quantitative real-time PCR)
290: maybe provide a bit more background as to the reason why the alu-qRT-PCR is preferred.
-we explained why Alu-PCR is used for detection of disseminated tumor cells
Here, human specific Alu-sequences, repetitive sequences interspersed over all chromosomes, are used to detect tumor cells of human origin. Using human specific alu quantitative real-time PCR (qPCR) …
Line 501: change "proofs" to "observations", and again a "the" missing before the "chicken embryo"
-done
line 509: add a reference which established these percentages.
-we added the following reference: C. H. Wong, K. W. Siah and A.W.Lo, Estimation of clinical trial success rates and related parameters, Biostatistics (2019) 20, 2, pp. 273–286 doi:10.1093/biostatistics/kxx069
7.2: one problem with the CAM model is that it is a closed system. Although the liver and kidney do function. Most excreted products go to the allantoic fluid, which is partly taken up again. If drugs, and especially small molecules, are not metabolized, they could re-enter the embryo. This might increas the toxicity of drugs. I'm not sure if this has been investigated, but it should be discussed. it is written as a benefit in 551, but obviously also bears a drawback.
It is true that the chicken embryo is an enclosed system and the metabolic processes of whichever kind of compounds might be different to that defined in mice and in human. Therefore, first the appropriate doses and a suitable administration regimen for the in ovo test of each compound must be defined, but we cannot directly transfer this system then into the mouse.
Due to its enclosed character, the doses adapted for in ovo test are frequently much lower than that used in mouse and in human. In this way, it could be also one benefit of the in ovo model, that reduced compound quantity is required (written in 575). Of course, after the validation of compound efficacy, it should be redefined the adequate dose for transfer into human applications.
We have rephrased the paragraph accordingly: Otherwise, if drugs, and especially small molecules, are not metabolized, they could re-enter the embryo and possibly increase the toxicity of drugs. Thus, after the validation of compound efficacy, the adequate dose for transfer into human applications must be re-defined.
Lines 585-599: it should be discussed how similar these organs (kidney, liver and gut) function in comparison to mammals. At least the kideny will have some differences.
We agree with the reviewer that the specific anatomy of the avian kidney leads to differences in a few kidney-related functions: the chicken excretes the nitrogenous waste as uric acid, in mammals urea is excreted. Since urinary bladder is lacking in the chicken, there is no urine storage capacity. https://doi.org/10.1152/advan.00031.2018
We rephrased this paragraph: But differently, the chicken excretes the nitrogenous waste as uric acid, in mammals urea is excreted. Since urinary bladder is lacking in the chicken, there is no urine stor-age capacity. [138]. Compared to body size, the avian liver is larger than in mammals, there is less connective tissue and no true lobular structure [139,140]. Although the metabolic processes in the liver of birds and mammals are similar, there are differ-ences, for example, in lipogenesis [139,141]. Regarding the excretion and resorption, the gut system of birds should also be considered for pharmacokinetics. Compared to the mammalian stomach, posteriorly to the proventriculus a second sacular, disc shaped and very muscular structure, the ventriculus or avian gizzard, is formed start-ing at ED6 [142,143]. The gizzard´s expression of proteins or receptors may have an in-fluence on the pharmacokinetics of the studied drug [144,145].minor:
line 45 change interview to review
done
line 137: explain the abbreviation FACS
done
line 483: Change "to answer these concerns, chicken embryo..." to "to answer these concerns, the CAM model..."
done
Re-read part 7 for grammer, often, the article "the" is lacking where needed: such as in line 489: the immune functionality in the chicken embryo...
We were correcting the english language

Reviewer 2 Report
The manuscript by Fischer et al. is an excellent review of the CAM essay in question-and-answer format. The topics chosen are very interesting and discussed in a very clear manner with an abundant and appropriate up-to-date literature review. This review will certainly be very useful to readers who are interested in this very useful in vivo model that is part of the animal welfare movement.
Author Response
we thank the reviewer for this positive reply.